# Peer review of "Physical Properties and Hydration Characteristics of Low-Heat Portland Cement at High-Altitude"

_materials, 2023, doi:10.3390/ma16083110_

Round 1

Reviewer 1 Report

General Comments:

The paper is well structured; it contains all key parts of the scientific article. In the text section the results are generally well presented, the quality of this part of the paper is good – authors focus sufficient on the both preparation and editing of the manuscript as well as the formulation of the presented statements. On the other hand, the quality of the figures is only moderate and it should be improved. In terms of originality the manuscript is devoted to determination of the mechanical properties and the hydration characteristics of low-heat Portland cement in a high-altitude environment, thus, this topic will be interesting for the Cement chemistry researches and Civil engineers. Scientific discussion of the content of the work is sufficient – authors cited 37 scientific and technical papers, most of them were published during the last decade. On the other hand, there are some of questions that authors need to answer in order to bring the paper to the level of a submission to Materials.

Specific Comments:

1.      The abstract should be rewritten and include all the essential information of the work, briefly mentioning the experimental conditions and analytical methods used. It is very vague at the moment. What the authors mean by "low-heat cement” or “high-altitude environment”. Highlighting the practical and theoretical significance would add additional value to your work. This part of the paper should clearly describe the core of the problem you are addressing, what you did, found and recommend to the readers. It will help a prospective reader of the abstract to decide if they wish to read the entire article.

2.      Introduction, Paragraph 1. The authors use 2 terms "Cement" and "Portland cement" as synonyms. These are not synonyms, as the latter is only one variety of cements. Please fix it. You can use the abbreviation – OPC

3.      Introduction, Paragraph 2. Incorrect information provided. There are other types of low-heat cements, not only those containing a high amount of belite. Please rethink and correct the first sentence.

4.      What do you mean by "from a microscopic point of view" (penultimate sentence). Please explain.

5.      Figure 1 and everywhere else. Why is it not marked which Title on the Y axis? In addition, the peaks are marked with numbers of very small dimensions and are difficult to see.

6.      The sum of the oxides in Table 1 is 99.74%. Mark the remaining 0.26% as Other.

7.      Page 3. I don't understand the first sentence. If there is no sand, it is a paste – why do you talk about mortar below. Please write more clearly. Also, remove the cube sign from the mm. I didn't quite understand why Figure 2 was needed. What makes it so special that you provide a photo of this camera?

8.      Page 4. If you mention other manufacturers, please also mention the manufacturer of the thermogravimetric analysis device.

9.      Page 4 and everywhere else – the unit of strength is MPa, not Mpa.

10.  Page 4, Test Methods, the last sentense. Maybe USA, not America

11.  Correct the title of section 3.2 – it should start with a capital letter. Same under Figure 9.

12.  Figure 10. Ca/Si ratio – what do you mean – molar ratio, mass ratio or something else? Please mention. Cite the literature that a Ca/Si ratio higher than 2 indicates the presence of portlandite in the sample. How many times were EDX measurements taken at the same point? If only one, the reliability of the results is questionable.

13.  The first conclusion presents the strength results of the samples. I would like to see a short theoretical explanation (perhaps in connection with other conclusions) why such dependencies are obtained.

I think that after a major revision including the suggested changes the paper could be accepted for publication in the Journal.

Reviewer 2 Report

I have several general concerns and several specific comments.

General Concerns

The manuscript suffers from:

1. Lack of organization. The manuscript jumps from Introduction to Materials and Methods. The manuscript need to have different sections for (a) Introduction and Background, (b) Literature Review, (c) Novelty and Objectives of the research, (d) Methodology, (e) Materials, (f) Test Matrix, Number of Specimens Tested for each Test in the Matrix, (g) Test Procedures, (h) Specimen Preparation, (i) Test Results, (j) Analysis of Results, and (k) Conclusions. Organized this way, the reader does not have to search for where each of these important topics are discussed.For example, Literature Review does not belong in the Introduction. Otherwise, the reader will be confused and eventually disinterested in reading further the manuscript. The manuscript SHOULD be reorganized as discussed in this comment.

2. English Language: The manuscript needs to be written with the help of an English speaking professional technical writer. 

3. The manuscript does not clearly state what is the these or objective of the research. Additionally, the manuscript does not discuss how these objectives will be achieved. The manuscript dives into the confusing details of the research in a disorganized way. The reader will not be interested to read further that the abstract.

Specific Comments

1. Abstract: There is hardly any introductory remarks in the abstract. The bulk of the Abstract is presentation of results. There is no mention of flexural strength. There are problems with sentences that are not complete sentences, awkward and unclear words, etc (Please see the highlighted in YELLOW in the authors' abstract below). The ABSTRACT needs to be expanded to include all the tests. Additionally, proper English needs to be used. Line 11 is not a sentence. Line 15 does not make sense.

_______________________________________________________________

Abstract: In this study, the mortars was prepared by using the low-heat cement and cured in the simulated high-altitude environment. The physical properties, hydration characteristics of the low-heat cement curing in simulated high-altitude environmentThe results showed that the high-altitude environment is beneficial for the compressive strength of low-heat Portland cement  mortars developed in early curing age, but it is not useful for the compressive strength development of low-heat Portland cement mortars in later curing age. The hydration products will dehydrate condition and the Ca/Si ratio trend decrease in low-pressure and variable temperature.

_______________________________________________________________

2. Line 58, What is the "..low carbon property..."? 

3. Lines 58 - 60, This is not a sentence.

4. Lines 60 - 63, Only hydration characteristics is mentioned. Where are the other properties.

5. Line 67, "In this Experiment" Is there only one experiment? Awkward English.

6. Figure 1 is very hard to read.

7. Line 71, What is the grain size distribution of the sand? 

8. Line 85, The authors jump into the discussion of the two different molds without introducing why these two mold sizes are used. First the test procedures needs to be discussed. Then the details will follow. There are many surprises like this in the manuscript.

9.Line 65,  Need to discuss and present a test matrix in a table form for clarity.

10. There is no mention in the manuscript on how many specimens were tested for each condition. This is very important.

11. Line 107, Shrinkage issue  appears from nowhere. There is no mention in the ABSTRACT nor in ant discussion preceding this line. Is shrinkage a mechanical property? 

12.Section 2.2 is Test Method. Section 2.2.2 is also Test Method. Confusing. 

13 Line 133, Which strength property? 

14. Lines 139 - 143, This belongs in the Literature Review section.

15. Lines 144 - 149, How did the authors show the trend? There is no mention of the process of demonstrating the trend. Was it done by just comparing Figures 3 and 4? There is a need for a more elaborate method.

16. Figure 6 is not easy to read.

17. Line 271, Not a sentence.

18. Line 273, What is "IT"?

19. Section 4.0, Conclusions, The conclusions do not mention the results of all the tests (Example SHRINKAGE). The conclusions from all the tests need to be summarized. Otherwise why do the tests if the results will not be part of the conclusions.

Reviewer 3 Report

Recommendations and questions for authors:

·         It will be necessary to add to the text of the paper the authors' motivation for the presented works. Are these the outputs of a research project? How will the authors proceed further in the presented area?

·         I recommend that the authors add to the paper a comparison of the Chinese testing standards used with other (world) standards.

·         I miss a more detailed description of the laboratory and measurement techniques used.

·         The graphs in Figures 7, 9, and 10 need to be edited. The text data in these graphs is very difficult to read.

·         How will the outputs be used in design practice and what will be the potential economic benefit?

·         Have the authors implemented other testing methods? The presented methods will clearly influence other physical properties of the samples.

Round 2

Reviewer 1 Report

The authors have done enough to correct the first version of their work. In their response, they explained well what they were focusing on. The authors have largely taken into account my comments and suggestions. The newly formulated Abstract and Conclusions well describe the main results obtained in the work. As I noted in the first review, the results obtained can have practical value. Hence, while appreciating that the results are generally well presented, quality of the paper is good – authors focus sufficient on the both preparation and editing of the manuscript as well as the formulation of the presented statements – in my opinion, at this level of readiness, the paper could be accepted for publication in the Materials after Minor Revision. Three remarks remain:

1.       I disagree with the answer to Point 6. The Y-axis of your Figure 1 must have the name Intensity, cps (counts per second). If multiple XRD curves are presented in the Figure, the name of the Y-axis should be Intensity, a.u. (adequate units) or r.u (relative units).

2.       Point 8. Once again I suggest to throw out Figure 2 from the paper. It does not provide any scientific information.

3.       Point 13: Figure 10. You didn't answer my question: “How many times were EDX measurements taken at the same point? If only one, the reliability of the results is questionable”.

Reviewer 2 Report

The authors have adequately responded to my comments.
